# Emergence and spread of SARS-CoV-2 lineage B.1.620 with variant of concern-like mutations and deletions

Distinct SARS-CoV-2 lineages, discovered through various genomic surveillance initiatives, have emerged during the pandemic following unprecedented reductions in worldwide human mobility. We here describe a SARS-CoV-2 lineage - designated B.1.620 - discovered in Lithuania and carrying many mutations and deletions in the spike protein shared with widespread variants of concern (VOCs), including E484K, S477N and deletions HV69Δ, Y144Δ, and LLA241/243Δ. As well as documenting the suite of mutations this lineage carries, we also describe its potential to be resistant to neutralising antibodies, accompanying travel histories for a subset of European cases, evidence of local B.1.620 transmission in Europe with a focus on Lithuania, and significance of its prevalence in Central Africa owing to recent genome sequencing efforts there. We make a case for its likely Central African origin using advanced phylogeographic inference methodologies incorporating recorded travel histories of infected travellers.

Over a year into the pandemic and with an unprecedented reduction in human mobility worldwide, distinct SARS-CoV-2 lineages have arisen in multiple geographic areas around the world[1–3]. New lineages are constantly appearing (and disappearing) all over the world and may be designated variant under investigation (VUI) if considered to have concerning epidemiological, immunological or pathogenic properties. So far, four lineages (i.e. B.1.1.7, B.1.351, P.1 and B.1.617.2 according to the Pango SARS-CoV-2 lineage nomenclature[4,5]) have been universally categorised as variants of concern (VOCs), due to evidence of increased transmissibility, disease severity and/or possible reduced vaccine efficacy. An even broader category termed variant of interest (VOI) encompasses lineages that are suspected to have an altered phenotype implied by their mutation profile.

In some cases, a lineage may rise to high frequency in one location and seed others in its vicinity, such as lineage B.1.177 that became prevalent in Spain and was later spread across the rest of Europe[2]. In others, reductions in human mobility, insufficient surveillance and passage of time allowed lineages to emerge and rise to high frequency in certain areas, as has happened with lineage A.23.1 in Uganda[6], a pattern reminiscent of holdover H1N1 lineages discovered in West Africa years after the 2009 pandemic[7]. In the absence of routine genomic surveillance at their origin location, diverged lineages may still be observed as travel cases or transmission chains sparked by such in countries that do have sequencing programmes in place. A unique SARS-CoV-2 variant found in Iran early in the pandemic was characterised in this way[8], and recently travellers returning from Tanzania were found to be infected with a lineage bearing multiple amino acid changes of concern[9]. As more countries launch their own SARS-CoV-2 sequencing programmes, introduced strains are easier to detect since they tend to be atypical of a host country's endemic SARS-CoV-2 diversity, particularly so when introduced lineages have accumulated genetic diversity not observed previously, a phenomenon that is characterised by long branches in phylogenetic trees. In Rwanda, this was exemplified by the detection of lineage B.1.380[6], which was characteristic of Rwandan and Ugandan epidemics at the time. The same sequencing programme was then perfectly positioned to observe a sweep where B.1.380 was replaced by lineage A.23.1[6], which was first detected in Uganda[10], and to detect the country's first cases of B.1.1.7 and B.1.351. Similarly, sequencing programmes in Europe were witness to the rapid displacement of pan-European and endemic lineages with VOCs, primarily B.1.1.7 (e.g. Lyngse et al.[11]).

Given the appearance of VOCs towards the end of 2020 and the continued detection of previously unobserved SARS-CoV-2 diversity, it stands to reason that more variants of interest (VOIs), and perhaps even VOCs, can and likely do circulate in areas of the world where access to genome sequencing is not available nor provided as a service by international organisations. Lineage A.23.1[10] from Uganda and a provisionally designated variant of interest A.VOI.V2[9] from Tanzania might represent the first detections of a much more diverse pool of variants circulating in Africa. We here describe a similar case in the form of a lineage designated B.1.620 that first caught our attention as a result of what was initially a small outbreak caused by a distinct and diverged lineage previously not detected in Lithuania, bearing multiple VOC-like mutations and deletions, many of which substantially alter the spike protein.

The first samples of B.1.620 in Lithuania were redirected to sequencing because they were flagged by occasional targeted PCR testing for SARS-CoV-2 spike protein mutation E484K repeated on PCR-positive samples. Starting April 2nd 2021, targeted E484K PCR confirmed a growing cluster of cases with this mutation in Anykščiai municipality in Utena county with a total

of 43 E484K$^+$ cases out of 81 tested by April 28th (Supplementary Fig. S1). Up to this point, the Lithuanian genomic surveillance programme had sequenced over 10% of PCR-positive SARS-CoV-2 cases in Lithuania and identified few lineages with E484K circulating in Lithuania. During initial B.1.620 circulation in Lithuania the only other E484K-bearing lineages in Lithuania had been B.1.351 (one isolated case in Kaunas county, and 12 cases from a transmission chain centred in Vilnius county) and B.1.1.318 (one isolated case in Alytus county), none of which had been found in Utena county despite a high epidemic sequencing coverage in Lithuania (Supplementary Fig. S2).

An in-depth search for relatives of this lineage on GISAID[12] uncovered a few genomes from Europe initially, though more continue to be found since B.1.620 received its Pango lineage designation which was subsequently integrated into GISAID. This lineage now includes genomes from a number of European countries such as France, Switzerland, Belgium, Germany, England, Scotland, Italy, Spain, Czechia, Norway, Sweden, Ireland, and Portugal, North America: the United States (US) and Canada, and most recently The Philippines and South Korea in Asia. Interestingly, a considerable proportion of initial European cases turned out to be travellers returning from Cameroon. Since late April 2021, sequencing teams operating in central Africa, primarily working on samples from the Central African Republic, Equatorial Guinea, the Democratic Republic of the Congo, Gabon and lately the Republic of Congo have been submitting B.1.620 genomes to GISAID.

We here describe the mutations and deletions the B.1.620 lineage carries, many of which were previously observed in individual VOCs, but not in combination, and present evidence that this lineage likely originated in central Africa and is likely to circulate in the wider region where its prevalence is expected to be high. By combining collected travel records from infected patients entering different European countries, and by exploiting this information in a recently developed Bayesian phylogeographic inference methodology[13,14], we reconstruct the dispersal of lineage B.1.620 from its inferred origin in the Central African Republic to several of its neighbouring countries, Europe and the US. Finally, we provide a description of local transmission in Lithuania, France, Spain, Italy, and Germany through phylogenetic and phylogeographic analysis, and in Belgium through the collection of travel records.

## Results

**B.1.620 carries numerous VOC mutations and deletions.** Lineage B.1.620 attracted our attention due to large numbers of unique mutations in B.1.620 genomes from Lithuania in nextclade analyses (its genomes are 18 mutations away from nearest relatives and 26 from reference strain Wuhan-Hu-1), and those genomes initially being assigned to clade 20A, corresponding to B.1 in Pangolin nomenclature[4,5]. Meanwhile, Pangolin (using the 2021-04-01 version of pangoLEARN) variously misclassified B.1.620 genomes as B.1.177 or B.1.177.57 and occasionally as correct but unhelpful B.1, prior to the official designation of B.1.620 by the Pango SARS-CoV-2 lineage nomenclature team. To this day even after official designation Pangolin still often struggles with B.1.620 sequences and classifies them as various VOCs (often as B.1.1.7) when not used in the new UShER mode and vice versa sometimes classifies non-B.1.620 genomes as B.1.620. Closer inspection of B.1.620 genomes revealed that this lineage carries a number of mutations and deletions that have been previously observed individually in VOCs and VOIs (Fig. 1 and Supplementary Fig. S3), but had not been seen in combination. Despite sharing multiple mutations and deletions with known VOCs (most prominently HV69/70Δ, LLA241/243Δ,

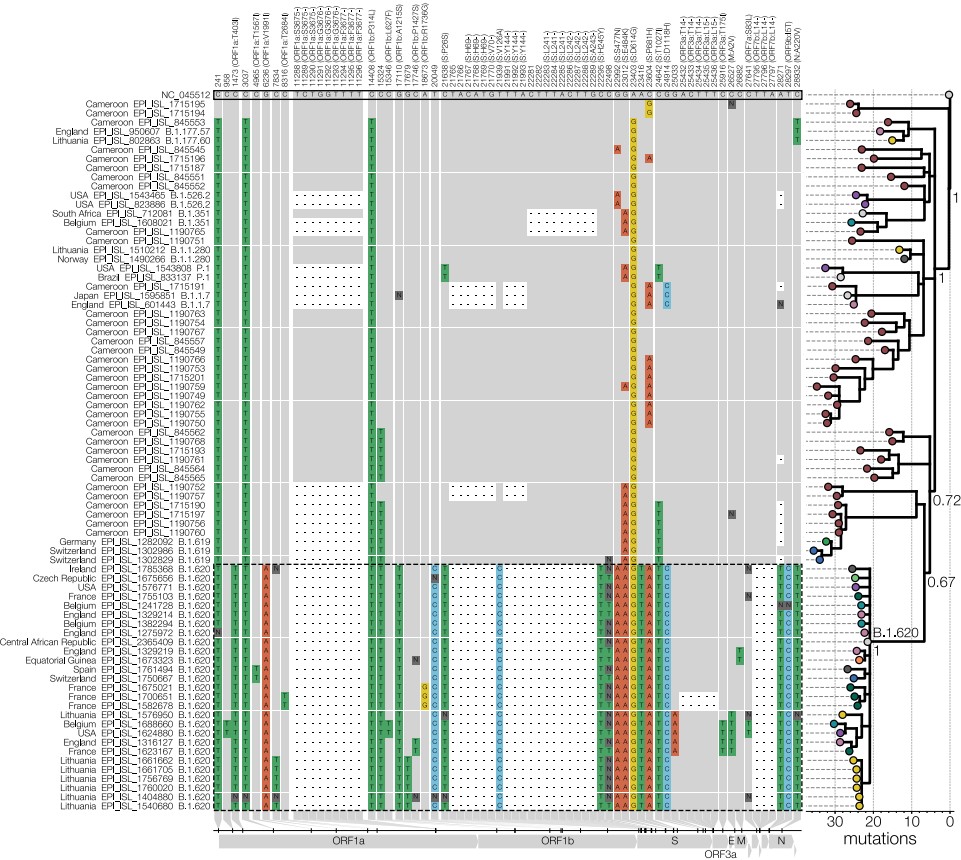

**Fig. 1 Lineage-defining SNPs of lineage B.1.620.** Only SNPs that differentiate B.1.620 (genomes outlined with a dashed line) from the reference (GenBank accession NC_045512) and that are shared by at least two B.1.620 genomes are shown in the condensed SNP alignment. Sites identical to the reference are shown in grey, changes from the reference are indicated and coloured by nucleotide (green for thymidine, red for adenosine, blue for cytosine, yellow for guanine, dark grey for ambiguities, black for gaps). The first 100 and the last 50 nucleotides are not included in the figure but were used to infer the phylogeny. If a mutation results in an amino acid change, the column label indicates the gene, reference amino acid, amino acid site, and amino acid change in brackets. The phylogeny (branch lengths in the number of mutations) on the right shows the relationships between depicted genomes and was rooted on the reference sequence with coloured circles at the tips indicating the country from which the genome came. Posterior probabilities of nodes leading up to lineage B.1.620 are shown near each node with the long branch leading to lineage B.1.620 labelled as 'B.1.620'.

S477N, E484K and P681H), lineage B.1.620 does not appear to be of recombinant origin (Supplementary Fig. S4).

Through travel-related cases of B.1.620 discussed later we suspected Cameroon as the immediate source of this lineage and therefore sought to identify close relatives of this lineage there. While genomic surveillance in Cameroon has been limited, the genomes that have been shared on GISAID are quite diverse and informative. A handful appears to bear several mutations in common with lineage B.1.620 and could be its distant relatives (Fig. 1). Synonymous mutations at site 15324 and S:T1027I appear to be some of the earliest mutations that occurred in the evolution of lineage B.1.620, both of which are found in at least one other lineage associated with Cameroon (B.1.619), followed by S:E484K which also appears in genomes closest to lineage B.1.620. Even though the closest genomes to B.1.620 were sequenced from samples collected in January and February, lineage B.1.620 has 23 changes (mutations and deletions) leading up to it compared to the reference. During this study, SARS-CoV-2 genomes collected in January-March 2021 from the Central African Republic were deposited on GISAID, but none of them resembles forebearer or sibling lineages to B.1.620.

**B.1.620 is likely to escape antibody-mediated immunity.** Like most currently circulating variants, B.1.620 carries the D614G mutation, which enhances infectivity of SARS-CoV-2, likely through enhanced interactions with the ACE2 receptor by promoting the up-conformation of the receptor-binding domain (RDB)[15]. Furthermore, B.1.620 contains P26S, HV69/70Δ, V126A, Y144Δ, LLA241/243Δ and H245Y in the N-terminal domain (NTD) of the spike protein. The individual V126A and H245Y substitutions are still largely uncharacterised to the best of our knowledge, but might be counterparts to the R246I substitution in B.1.351, and the latter may interfere with a putative glycan binding pocket in the NTD[16]. All other mutations of B.1.620 in the NTD result in partial loss of neutralisation of convalescent serum and NTD-directed monoclonal antibodies[17]. This indicates that these mutations present in B.1.620 may have arisen as an escape to antibody-mediated immunity[18]. The spike protein of B.1.620 also carries both S477N and E484K mutations in the RBD, but in contrast to other VOCs not the N501Y or K417 mutations. Like the mutations in the NTD, S477N and E484K individually enable broad escape from antibody-mediated immunity[18]. Moreover, deep mutational scanning experiments have shown that these substitutions also increase the affinity of the RBD for the ACE2 receptor[19]. Both S477N and E484K occur on the same flexible loop at the periphery of the RDB-ACE2 interface[20].

We have modelled the RBD–ACE2 interface with the S477N and E484K substitutions using refinement in HADDOCK 2.4[21]. These models show that both individual substitutions and their combination produce a favourable interaction with comparable scores and individual energy terms to the ancestral RBD

(Supplementary Fig. S5). Whereas S477N may modulate the loop conformation[22], E484K may introduce new salt bridges with E35/E75 of ACE2. These results indicate that B.1.620 may escape antibody-mediated immunity while maintaining a favourable interaction with ACE2. The remaining mutations in the spike protein—P681H, T1027I and D1118H—are uncharacterised to the best of our knowledge. Of these, P681H is also located on the outer surface of the spike protein, directly preceding the multibasic S1/S2 furin cleavage site[23]. In contrast, T1027I and D1118H are both buried in the trimerisation interface of the S2 subunit[24].

While only limited empirical data are available, they seem to agree with the expectation that B.1.620 is likely to be antigenically drifted relative to primary genotypes. A report presented to the Lithuanian government on May 22, 2021[25] indicated that amongst 101 sequenced B.1.620 cases at the time, 13 were infections in fully vaccinated individuals, five of whom were younger than 57 years old. Though not systematised properly, sequencing indications for a substantial number of SARS-CoV-2 genomes from Lithuania were available, of which 213 were 'positive PCR at least 2 weeks after the second dose of vaccine', of which 195 were B.1.1.7 and 12 were B.1.620. Since detection of the first B.1.620 case on March 15, 2021, in Lithuania ~10,000 SARS-CoV-2 genomes were sequenced to date, 9251 of which were B.1.1.7 and 248 of which were B.1.620. Thus B.1.620 is found 2.4 times more often in vaccine breakthrough cases compared to its population prevalence, whereas for B.1.1.7 this enrichment is only 1.05-fold. Similarly, the frequency of B.1.620 across the five most affected European countries (Lithuania, Germany, Switzerland, France and Belgium) appears relatively stable though at a low level, unlike B.1.1.7 which has been in noticeable decline since April–May (Supplementary Fig. S6), presumably on account of increasing vaccination rates and improving weather in Europe.

**Local transmission of B.1.620 in Europe**. Local transmission of B.1.620 in Lithuania has been established as a result of monitoring the outbreak in Anykščiai municipality (Utena county, Lithuania) via sequencing and repeat PCR testing of SARS-CoV-2 positive samples for the presence of E484K and N501Y mutations, as well as looking for S gene target failure (SGTF) caused by the HV69Δ deletion. Genotypes identical to those found initially in Vilnius and Utena counties were later identified by sequencing in Panevėžys and Šiauliai counties, indicating continued transmission of lineage B.1.620 in-country. Interestingly, a single case in Tauragė county, Lithuania, identified by sequencing was a traveller returning from France found to be infected with a different genotype than the main outbreak lineage in Lithuania without evidence of onward transmission via local contact tracing efforts or genomic surveillance.

In addition to an ongoing disseminated outbreak of B.1.620 in Lithuania, genomes of this lineage have been found elsewhere in Europe. Though derived from separate introductions from the one that sparked outbreaks in Lithuania, other B.1.620 genomes from Europe appear to indicate ongoing transmission in Europe, with the clearest evidence of this in Germany and France, where emerging clades are comprised of identical or nearly identical genotypes (Fig. 2). Presenting evidence for local transmission in Europe, B.1.620 genomes from countries like Spain and Belgium (also see next section) were notably picked up by baseline surveillance and thus are likely to represent local circulation, though presumably at much lower levels at the time of writing. Figure 2 shows the aforementioned local transmission clusters in Lithuania, Spain (Vilassar De Mar, province of Barcelona), France (see below), and Germany (state of Bavaria), amongst numerous others.

In France, nine B.1.620 genomes (EPI ISL 1789089 - EPI ISL 1789097) were recently obtained from a large contact tracing

investigation of a single transmission chain. These infections in the municipality of Pontoise (Val d'Oise department, to the northwest of Paris) occurred in adults (ages 24–38) who were all asymptomatic at the time of sampling. Additional infections in Pontoise outside of this cluster occurred in four adults (ages 29–57) and form a monophyletic cluster with the other nine infected individuals (Supplementary Fig. S4). The putative index case for these infections has yet to be determined through contact tracing at the time of writing but these cases clearly point to the B.1.620 lineage circulating in the Val d'Oise department. These infections seem to stem from local ongoing transmission in the Île-de-France region, clustering with two patients ages 1 (sample from a children's hospital in Paris: Hôpital Necker-Enfants malades) and 69. These infections in Île-de-France in turn cluster with two infections from Le Havre (region of Normandy; 180km from Pontoise), pointing to either a travel event from Normandy to Île-de-France or possible local transmission in the north of France (Supplementary Fig. S4).

**B.1.620 likely circulates at high frequency in central Africa**. In the absence of routine surveillance at a location, sequencing infected travellers originating from there constitutes the next most efficient way to monitor distinct viral populations. This has been used successfully to uncover cryptic outbreaks of Zika virus in Cuba[26] and SARS-CoV-2 in Iran at the beginning of the pandemic[13]. The latter study describes a novel approach to accommodate differences in sampling location and location of infection, and is hence specifically targeted to exploit recorded travel histories of infected individuals in Bayesian phylogeographic inference, rather than arbitrarily assigning the origin of the sample to either location. When we first compiled our B.1.620 genomes dataset we had seven genomes from travellers and six were sampled in the Central African Republic (CAR) near the border with Cameroon, indicating the most plausible geographic region where B.1.620 is circulating widely to be central Africa (Supplementary Fig. S7). Neighbours of countries reporting local B.1.620 circulation (Cameroon, CAR, DRC, Gabon, Equatorial Guinea, and later the Republic of Congo) have either not submitted genomes to GISAID during the study period (Chad, Sudan, South Sudan, Burundi) or have epidemics dominated by SARS-CoV-2 lineages that are not B.1.620 (Supplementary Fig. S8).

The collected individual travel histories themselves point to several independent introductions of B.1.620 into Europe, with documented cases of infected travellers returning from Cameroon to Belgium, France and Switzerland, and from Mali to Czechia (Fig. 3). We note that the metadata for a returning traveller from Cameroon to Belgium (EPI_ISL_1498300) presents evidence of ongoing local transmission within Belgium of B.1.620. Whereas this patient had spent time in Cameroon from the 16th of January until the 7th of February, a positive sample was only collected on the 15th of March, 2021. Even when assuming a lengthy infectious period of up to twenty days[27], this patient's infection can not stem from his prior travel to Cameroon, which indicates an infection with B.1.620 within Belgium and hence stemming from contact within the patient's community. Additionally, two Belgian patients (EPI_ISL_1688635 and EPI_ISL_1688660) were likely infected by the former's niece who had travelled with her family to Cameroon and tested positive upon their return to Belgium. These findings are reinforced by more recent samples from Belgium, for which no travel history could be recorded and the patients declared not having left the country.

Using a Bayesian phylogeographic inference methodology that accommodates individual travel histories we were able to reconstruct location-annotated phylogenies at both the continent and country

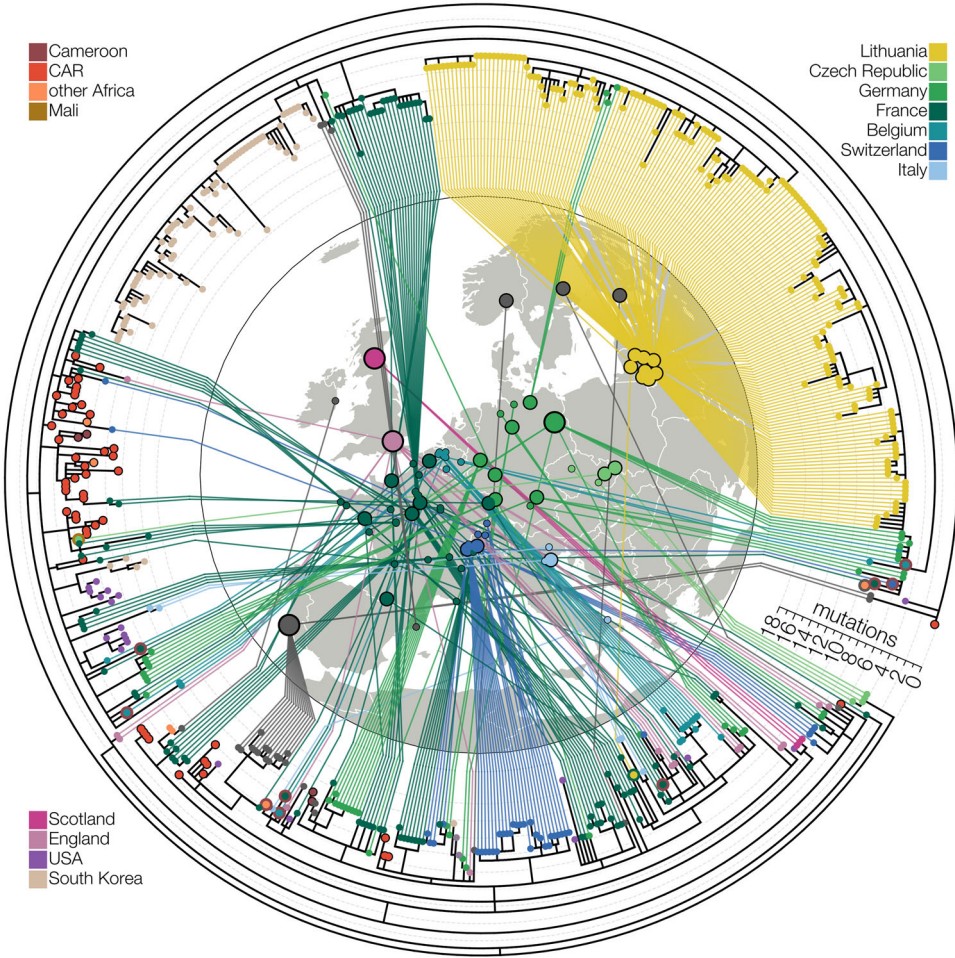

**Fig. 2 Maximum-likelihood tree of lineage B.1.620 in Europe.** Relationships between B.1.620 genomes, coloured by country of origin (same as Fig. 1) with a thicker coloured outline indicating the country of origin for travel cases. At least ten genomes shown (samples collected in Belgium, Switzerland, France and Equatorial Guinea) are from individuals who returned from Cameroon, one is from a traveller returning from Mali and one Lithuanian case returned from France. Genomes from the Central African Republic (CAR) and Czechia (returning traveller from Mali) are descended from the original B.1.620 genotype, while the genome from Equatorial Guinea is already closely related to genomes found in Europe and happens to be a travel case from Cameroon. Each genome is connected to the available geographic location in Europe with the smallest circles indicating municipality-level precision, intermediate size corresponding to county-level information (centred on county capital) and largest circle sizes indicating country-level information (centred on country capital). Countries are assigned the same colours as in Figs. 1 and 3.

levels. Figure 4A shows the MCC tree of the continent-level phylogeographic analysis, which yields 99.5% posterior support for an African origin of lineage B.1.620. From this inferred African origin, the variant then spread to different European countries via multiple introductions, which is confirmed by our collection of travel history records for individuals returning to these countries. Subsequent country-level phylogeographic analysis—shown in Fig. 4B—points to central Africa as the likely origin of this lineage, with the Central African Republic receiving posterior support of 80.5% and Cameroon 16.8%, taking up 97.3% of the probability mass together. Assuming a Central African Republic origin, the variant is estimated to have spread to Europe via a series of introductions, confirming what was also observed in our recorded travel history records. Interestingly, a single Lithuanian case—a returning traveller from France—does not cluster with the cluster of remaining sequences from Lithuania, illustrative of at least two independent introductions of lineage B.1.620 into Lithuania. Figure 4B also shows multiple separate B.1.620 introduction events from central Africa into the United Kingdom and the United States.

Air passenger flux out of Cameroon and Central African Republic (Fig. 5) shows that many travellers had African

countries as their destination, including many that have not reported any B.1.620 genomes to date. This suggests that B.1.620 could be circulating more widely in Africa and its detection in Europe has mostly occurred in countries with recent active genomic surveillance programmes. Detections of B.1.620 in African states neighbouring Cameroon and Central African Republic (Equatorial Guinea, Gabon, DRC and lately the Republic of Congo), even at low sequencing levels, suggest that B.1.620 may be prevalent in central Africa. We find this apparent rise to high frequency and rapid spread across large areas of Africa noteworthy in light of other findings reported here, namely that currently available B.1.620 genomes appeared suddenly in February 2021 (Fig. 3), are genetically homogeneous (Fig. 2), and to date have no clear close relatives (Fig. 1).

## Discussion
In this study, we have presented evidence that a SARS-CoV-2 lineage designated B.1.620, first detected in Europe in late February, is associated with the central African region, where it appears to circulate at high prevalence, and has been introduced

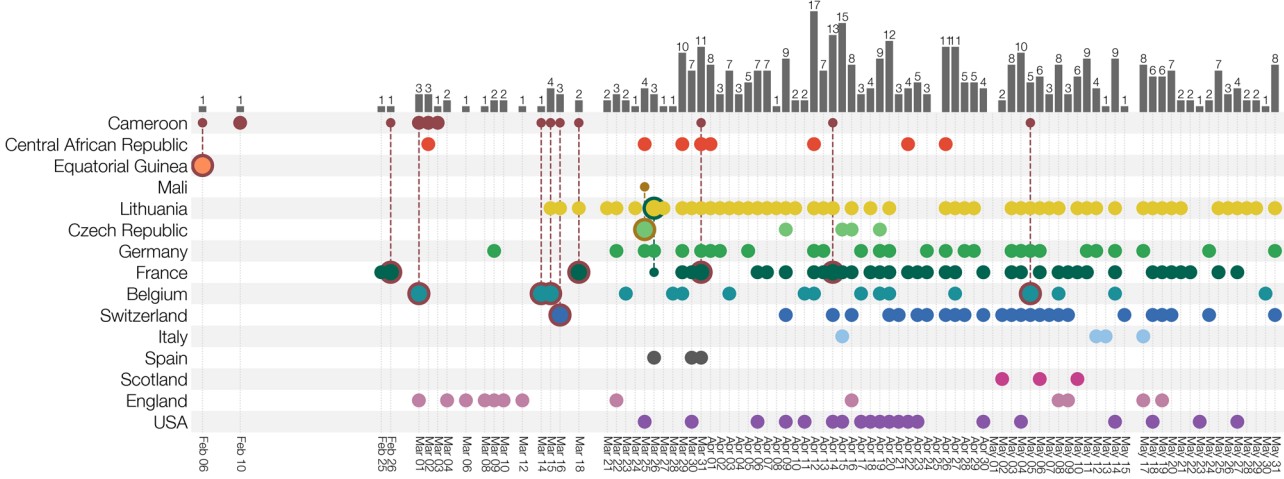

**Fig. 3 Known locations and travel history of B.1.620 cases.** Collection dates of B.1.620 genomes are shown for each country (rows). Genomes from travellers are outlined with colour indicating travel of origin (e.g. dark red for Cameroon) and connected to a smaller dot indicating which country's diversity is being sampled at the travel destination. Bars at the top indicate the number of genomes of B.1.620 available for a given date across all countries. Countries are assigned the same colours as in Fig. 1.

into Europe, North America, and Asia on multiple occasions. A fair number of known B.1.620 genomes that were sequenced in Europe stem from travel-related cases returning from Cameroon (Fig. 3), and recently sequenced genomes from CAR and Cameroon similarly belong to lineage B.1.620, suggesting that the central African region is likely to be the immediate source of this lineage. Importantly, our findings are quite insensitive to the actual sequence data used. Older datasets we used dating from the end of April 2021 (Supplementary Fig. S9) included only six genomes from CAR and travel cases in Europe coming from Cameroon and yet still confidently identified Cameroon as the immediate origin of lineage B.1.620. Adding more data from CAR (Supplementary Fig. S10) made available later made the Central African Republic the more likely country where B.1.620 circulated prior to spreading elsewhere, but ultimately no country other than CAR and Cameroon are considered as remotely plausible by the model.

Substantially higher passenger flux out of Cameroon compared to CAR (practically an order of magnitude) is a likely explanation for why B.1.620-infected travellers were overwhelmingly coming to Europe from Cameroon. So far the only observation that is difficult to explain is the Czech case returning from Mali, since Mali is over 1000 km away from Cameroon. We consider the introduction of B.1.620 from central Africa to Mali via land routes improbable, since outbreaks caused by B.1.620 have not been observed in Niger and Nigeria, the countries separating the region from Mali. The lack of any B.1.620 genomes from Nigeria in particular, one of the leaders in SARS-CoV-2 genome sequencing on the continent to date, despite higher civil air passenger volumes (Fig. 5) suggests other means of long-distance travel between central Africa and Mali[28,29].

In addition to the multiple introductions of the B.1.620 lineage we observe (Fig. 3) and estimate (Fig. 4) in Europe and North America, we also found evidence of local transmission of this lineage in Europe, with clearest evidence in Lithuania (Supplementary Fig. S1) followed by Germany and France (Fig. 3), and finally, Belgium and Catalonia, where B.1.620 genomes were picked up by baseline surveillance and infected individuals did not report having travelled abroad. B.1.620 is worrying for several reasons—its genomes are genetically homogeneous—as it appeared suddenly in February 2021 bearing a large number of VOC-like mutations and deletions in common with multiple VOCs (Supplementary Fig. S3), yet in the absence of any clear

close relatives or sampled antecedents (Fig. 1). The discovery of a novel lineage bearing many mutations of concern and with indications that they are introduced from locations where sequencing is not routine, is concerning and such occurrences may become an alarming norm.

The continued lack of genomic surveillance in multiple areas of the world, let alone equitable access to vaccines to drive transmission down, will continue to undermine efforts to control SARS-CoV-2 everywhere. Without the ability to identify unusual variants, to observe their evolution and learn from it, and to evaluate how vaccine-induced immunity protects against them, any response enacted by individual countries is reactive and, much like the process of evolution that generates variants of concern, short-sighted. The emergence of B.1.1.7 was unprecedented and has had a devastating impact on the state of the pandemic, so it is concerning that similar information gaps in global genomic surveillance still persist to this day. As an example we have shown that B.1.620 lacks intermediate relatives, resulting in a long branch that connects this lineage to the ancestral genotype of B.1. This could be the result of gradual but unsampled evolution, perhaps even far away from central Africa, but it could have also happened due to unusual selection pressures in immunosuppressed individuals[30] which is hypothesised for lineage B.1.1.7. The long branch leading to B.1.620 also means that we can not reconstruct the order of mutations that have occurred during the genesis of this lineage and therefore whether some amino acid changes have allowed others to happen by altering the fitness landscape via epistatic interactions[31]. Given the number of VOC-like mutations B.1.620 has, this is a significant loss.

Our work highlights that global inequalities, as far as infectious disease monitoring is concerned, have tangible impacts around the world and that until the SARS-CoV-2 pandemic is brought to heel everywhere, nowhere is safe for long. Additionally, we highlight the importance of collecting and sharing associated metadata with genome sequences, in particular regarding individual travel histories, as well as collection dates and locations, all of which are important to perform detailed phylogenetic and phylogeographic analysis. We only observed one single instance where a GISAID entry was accompanied by travel information and had to request such information for all the samples in our core dataset by contacting each individual lab. Whereas many labs were quick to provide the requested information, we were certainly not able to retrieve all related individual travel histories.

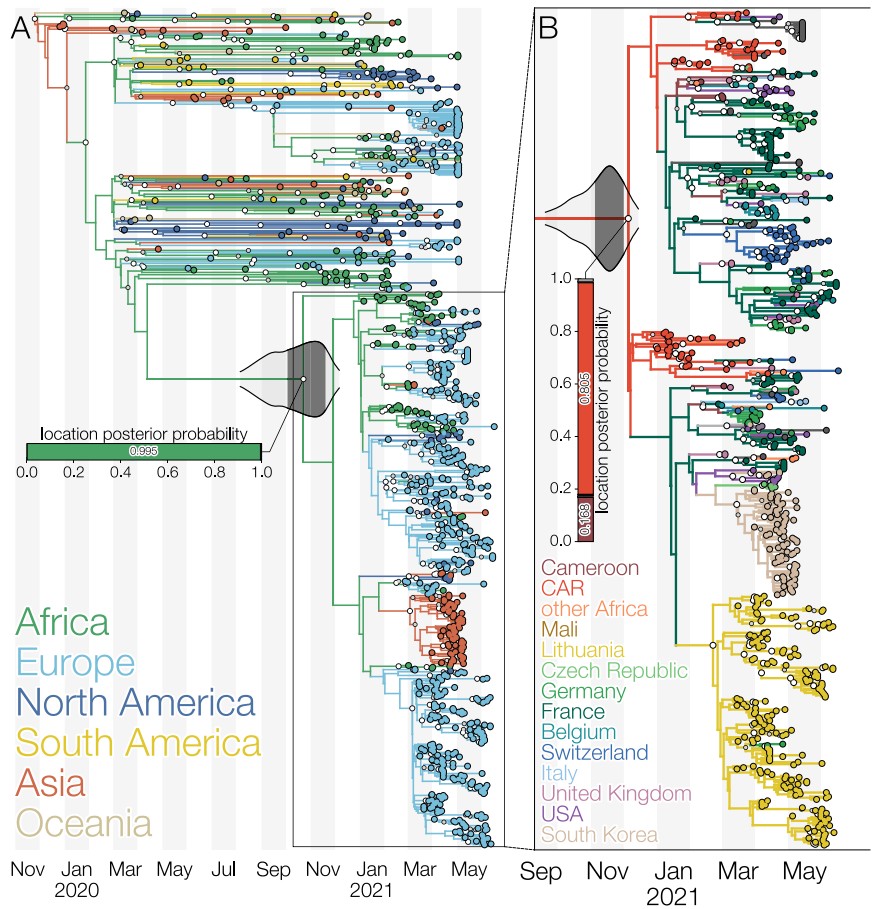

**Fig. 4 Maximum clade credibility trees of lineage B.1.620 coloured by reconstructed location using the latest available data as of June 2021. A** Global phylogeny of SARS-CoV-2 genomes with branches coloured by inferred continent from a Bayesian phylogeographic analysis that makes use of individual travel histories. Lineage B.1.620 is outlined and a horizontal bar shows the posterior probability of its common ancestor existing in a given continent. Africa is reconstructed as the most likely location (posterior probability 0.995) where B.1.620 originated. The 95% and 50% highest posterior density (HPD) intervals for the most recent common ancestor date of lineage B.1.620 are indicated with violin plots centred on the common ancestor. **B** Phylogeny of lineage B.1.620 with branches coloured by inferred country from a Bayesian phylogeographic analysis that makes use of travel histories. A vertical bar shows posterior probabilities of where the common ancestor of B.1.620 existed. In this analysis, Central African Republic (CAR) and Cameroon are reconstructed as the most likely locations (with posterior probabilities of 0.805 and 0.168, respectively) of the common ancestor of lineage B.1.620. Larger white dots at nodes indicate nodes with a posterior probability of at least 95%, while smaller grey circles indicate nodes with a posterior probability of at least 50%. The 95% and 50% highest posterior density (HPD) intervals for the most recent common ancestor date of lineage B.1.620 are indicated with violin plots centred on the common ancestor.

The scientific community therefore still faces the important task of reporting and sharing such critical metadata in a consistent manner, an aspect that has been brought to attention again during the ongoing pandemic[32,33].

## Methods

**Study design**. This study was initiated upon detection of SARS-CoV-2 strains in Lithuania bearing spike protein amino acid substitutions E484K, S477N and numerous B.1.1.7-like (HV69/70Δ and Y144Δ) and B.1.351-like (LLA241/243Δ) deletions, amongst others. In Lithuania, repeat PCR testing of SARS-CoV-2 positive samples is occasionally carried out to detect N501Y, E484K and S gene target failure (SGTF) caused by the HV69Δ deletion. Upon detection of E484K-positive cases, samples were redirected to sequencing. Initially identified cases of B.1.620 were mistakenly classified by pangolin as B.1.177 or B.1.177.57, while nextclade[34] assigned it to clade 20A rather than the expected 20E (EU1), while highlighting that B.1.620 sequences bore many unique mutations compared to the closest sequence. Searching GISAID for mutations E484K, S477N and HV69/70Δ, which are found in numerous VOCs individually but not in combination, identified additional genomes that contained other mutations and deletions found in B.1.620.

We downloaded all available sequences of this lineage from GISAID in July 2021, and identified members that clearly belonged to this lineage. Prior to official lineage designation as B.1.620, most of its genomes could be identified by the presence of spike protein E484K and S477N mutations and the HV69/70Δ deletion. Some of B.1.620 genomes were excluded from phylogenetic analyses because they

were misassembled (e.g. hCoV-19/Belgium/UZA-UA-24912930/2021 is missing deletions characteristic of this lineage but has the mutations) or had too many ambiguous sites (e.g. hCoV-19/France/ARA-HCL021061598501/2021) but we recovered travel information about them regardless as this may prove useful to perform travel history-aware phylogeographic reconstruction[13].

**SARS-CoV-2 whole-genome sequencing**. Every sample that tests positive for SARS-CoV-2 by PCR in Lithuania with Ct values < 30 may be redirected by the National Public Health Surveillance Laboratory to be sequenced by the European Centre for Disease Prevention and Control (ECDC), Vilnius University Hospital Santaros Klinikos (VUHSK), Hospital of Lithuanian University of Health Sciences Kauno Klinikos (HLUHSKK), Vilnius University Life Sciences Centre (VULSC) or Lithuanian University of Health Sciences (LUHS). Samples of this particular lineage were sequenced by ECDC using in-house protocols, infrastructure and assembly methods, VUHSK using Illumina COVIDSeq reagents, Illumina MiSeq platform, and assembled with covid-19-signal[35], HLUHSKC using Twist SARS-CoV-2 Research Panel reagents, Illumina NextSeq550 platform, and assembled with V-pipe[36], LUHS using ARTIC protocol, Oxford Nanopore Technologies MinION platform, and assembled using ARTIC bioinformatics protocol for SARS-CoV-2, and VULSC using ARTIC V3 protocol combined with Invitrogen Collibri reagents, Illumina MiniSeq platform, Illumina DRAGEN COVID Lineage combined with an in-house BLAST v2.10.18-based assembly protocol. Samples from CAR were sequenced using the very same ARTIC V3 protocol as the Lithuanian University of Health Sciences (LUHS).

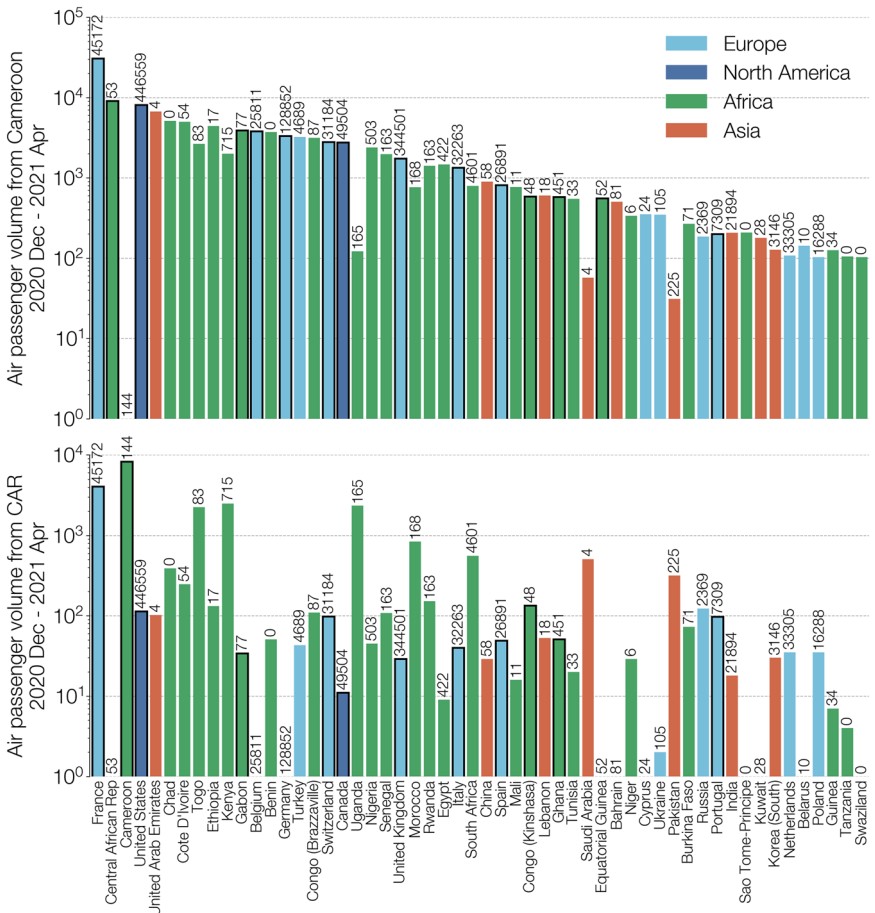

**Fig. 5 Total air passenger flows out of Cameroon (top) and Central African Republic (bottom) between December 2020 and April 2021.** Destination countries are sorted by total passenger volume arriving from Cameroon and Central African Republic (CAR) combined, coloured by continent (Europe in light blue, North America in dark blue, Africa in green, and Asia in red) and limited to countries where at least 100 passengers have arrived from either Cameroon or CAR between December 2020 and April 2021. Note the nearly order of magnitude greater passenger flux out of Cameroon compared to the Central African Republic (CAR). Numbers above each country's bar indicate the total number of genomes on GISAID from that country since January 1st 2021, according to GISAID's 2021-07-02 metadata release. Bars outlined in black represent countries that have submitted at least one B.1.620 genome as of June 2021.

All SARS-CoV-2 genomes used here were downloaded from GISAID. A GISAID acknowledgement table containing all genome accession numbers is included with this study as Supplementary Data 1.

**Associated travel history**. When available on GISAID as part of the uploaded metadata, we made use of this associated metadata information and contacted the submitting labs to determine precise travel dates. For all other cases, we retrieved individual travel histories by contacting the submitting labs—who then, in turn, contacted either the originating lab or the patient's general practitioner—for any travel records they may have available. This resulted in travel itineraries for 10 patients, with 7 of these also containing detailed dates for the recorded travel. When a returning traveller visited multiple countries on the return trip, we included all visited countries as possible locations of infection by using an ambiguity code in the phylogeographic analysis[13,14]. The travel history information collected can be found in Supplementary Table S1. While we were able to retrieve travel history for a fair number of cases, this information is considered private information in certain countries and we were hence unable to retrieve such data for a subset of our sequences.

SARS-CoV-2 genomes from the United Kingdom (UK) make up a sizeable proportion of any phylogenetic and phylogeographic analysis, given significant sequencing efforts by the COVID-19 Genomics UK Consortium. Given the lack of individual travel histories for B.1.620 genomes from England in our dataset, we investigated the passenger volumes from all airports in Cameroon and the Central African Republic to all airports internationally, incorporating volumes from both direct and connecting flights between December 2020 and April 2021, from the International Air Transportation Association (IATA[37]). These passenger data cover the time frame of our estimated B.1.620 lineage since its origin (see 'Results' section), with the passenger volumes for February having become available at the time of writing as these data need to be retrieved and processed. These air

passenger flux data reveal a very real possibility of missing travel histories from Cameroon for B.1.620 cases in England, given that over 98% (i.e. 852 out of 867) of the passengers from Cameroon to the UK during this time frame had an English airport (London, Manchester or Birmingham) as their final destination. At the time of writing, information on the origin of B.1.620 infections detected in England is not available.

**Modelling RBD–ACE2 interaction**. We have modelled the RBD–ACE2 interface with the S477N and E484K substitutions using the final refinement step of HADDOCK 2.4[21]. We used the crystal structure of ACE2 (19-615) bound to SARS-CoV-2 RBD (PDB ID: 6m0j[20]) as a starting point and introduced the substitutions using UCSF ChimeraX[38]. We used default parameters for refinement with extended molecular dynamics (MD) simulation (steps for heating phase: 200, steps for 300K phase: 2500, steps for cooling phase: 1000).

**Phylogenetic and phylogeographic analysis**. We combined 614 sequences belonging to lineage B.1.620 with sequences from lineages that have circulated in Lithuania at appreciable levels: B.1.1.7, B.1.1.280, B.1.177.60 and other VOCs that share mutations with lineage B.1.620: B.1.351, P.1 and B.1.526.2. We included high-quality sequences from Cameroon that were closest to lineage B.1.620 as well as the reference SARS-CoV-2 genome NC_045512. Some sequences had clusters of SNPs different from the reference at the ends of the genome, particularly the 5′ end. In such cases, the ends of the genomes were trimmed to exclude these regions of likely sequencing or assembly error. This resulted in a core set of 665 genomes, which is visualised in Supplementary Fig. S4, that serves as the starting point for our phylogenetic and phylogeographic analyses. This core set was subsequently combined with 250 randomly selected sequences from the Nextstrain global analysis on April 29, 2021 (https://nextstrain.org/ncov/global[34]) to provide context for the B.1.620 analysis, plus an additional two reference sequences: Wuhan/Hu-1/

2019 and Wuhan/WH01/2019. We filtered these sequences based on metadata completeness and added an additional four Chinese sequences as well as eight non-Chinese sequences from Asia spanning both A and B lineages, in order to balance the representation of different continents in our analyses. These sequences were aligned in MAFFT (FFT-NS-2 setting)[39] with insertions relative to reference removed, and 5′ and 3′ untranslated regions of the genome that were susceptible to sequencing and assembly error trimmed. We employed TempEst[40] to inspect the dataset for any data quality issues that could result in an excess or shortage of private mutations in any sequences, or would point to assembly or any other type of sequencing issues.

To look for sequences that could resolve the long period of unobserved evolution separating lineage B.1.620 from its closest relatives, we constructed a BLAST nucleotide database[41] of all contemporary SARS-CoV-2 lineages available via GISAID (accessed 2021-07-01, $n = 2,038,838$). We queried this database using a synthetic B.1.620-like sequence containing SNPs and deletions shared by B.1.620 sequences England/CAMC-13B04C1/2021 and France/PDL-IPP07069/2021 that were not present in the reference sequence Wuhan/Hu/2019. The synthetic query sequence was primarily comprised of ambiguous nucleotides (N) except for 100 nt surrounding each mutation or deletion characteristic of B.1.620. We checked the top 500 matches to see if the mutations they carry, their pango lineages or phylogenetic placement via IQ-TREE[42]—using a general time-reversible substitution model with among-site rate variation (GTR+$\Gamma_4$[43,44])—could identify sequences closer to lineage B.1.620 than B.1.619. No such sequences were identified.

We performed Bayesian model selection through (log) marginal likelihood estimation to determine the combination of substitution, molecular clock and coalescent models that best fits the data. To this end, we employed generalised stepping-stone sampling (GSS)[45] by running an initial Markov chain of 5 million iterations, followed by 50 path steps that each comprise 100,000 iterations, sampling every 500th iteration. We found that a combination of a non-parametric skygrid coalescent model[46], an uncorrelated relaxed clock model with underlying lognormal distribution[47] and a GTR+$\Gamma_4$ substitution model provided the optimal model fit to the data. We employed Hamiltonian Monte Carlo sampling to efficiently infer the skygrid's parameters[48].

We subsequently performed a discrete Bayesian phylogeographic analysis in BEAST 1.10.5[49] using a recently developed model that is able to incorporate available individual travel history information associated with the collected samples[13,14]. Exploiting such information can yield more realistic reconstructions of virus spread, particularly when travellers from unsampled or under-sampled locations are included to mitigate sampling bias. When the travel date for a sample could not be retrieved, we treated the time when the traveller started the journey as a random variable, and specified normal prior distributions over these random variables informed by an estimate of time of infection and truncated to be positive (back-in-time) relative to sampling date. As in previous work[13,14], we used a mean of 10 days before sampling based on a mean incubation time of 5 days[50], a constant ascertainment period of 5 days between symptom onset and testing[51], and a standard deviation of 3 days to incorporate the uncertainty on the incubation time.

In our phylogeographic analysis, we made use of Bayesian stochastic search variable selection (BSSVS) to simultaneously determine which migration rates are zero depending on the evidence in the data and infer ancestral locations, in addition to providing a Bayes factor test to identify migration rates of significance[52]. We first performed a continent-level phylogeographic analysis by aggregating sampling locations as well as the individual travel histories that occurred between continents. To ensure consistent spatial reconstruction regardless of sampling, we fixed the root location of this tree to be in Asia—so as to match the known epidemiology of the COVID-19 pandemic. Conditional on the results of this analysis, we performed a country-level analysis on the B.1.620 lineage and its parental lineage, in order to substantially reduce the computational burden and statistical complexity associated with having 87 sampling locations in a travel history-aware phylogeographic analysis. We made use of the following prior specifications for this analysis: a gamma (shape = 0.001; scale = 1000) prior on the skygrid precision parameter, Dirichlet (1.0, $K$) priors on all sets of frequencies (with $K$ the number of categories), Gamma prior distributions (shape = rate = 1.0) on the unnormalized transition rates between locations[52], a Poisson prior (country level: $\lambda = 28$; continent level: $\lambda = 5$) on the sum of non-zero transition rates between locations, a CTMC reference prior on the mean evolutionary rate and as well as on the overall (constant) diffusion rate[53]. In the country-level analysis, we assumed a normally distributed root height prior on the time of origin of B.1.620's parental lineage, with a mean on the 27th of February 2020 and standard deviation of 2 weeks, as derived from the corresponding internal node's 95% highest posterior density interval in the preceding continent-level analysis. For continent-level analysis 18 independent Markov chains were set up, running for ~50 million states and sampling every 40,000th state. All 18 runs were then combined after removing 10% of the states as burnin, giving a total MCMC length of 810 million states. For country-level analysis 16 independent Markov chains were set up, running for ~3.5 million states and also sampling every 40,000th state. All 16 runs were then combined after removing 10% of the states as burnin, giving a total MCMC length of 50.4 million states. Both continent-level and country-level combined run were inspected using Tracer v1.7[54] to confirm that effective sample sizes (ESSs) for all relevant parameters were at least 200. We used TreeAnnotator to

construct maximum clade credibility (MCC) trees for both posterior sets of trees and used baltic (https://github.com/evogytis/baltic) to visualise it.

In addition to sophisticated phylogeographic analyses, we also depict the raw relationships between SARS-CoV-2 in the core dataset of 665 genomes using substitution phylogenies. Figure 2 and Supplementary Fig. S4 depict maximum-likelihood phylogenies inferred from the core dataset using PhyML[55] under the HKY+$\Gamma_4$ model of nucleotide substitution[44,56] which was then rooted on the reference sequence. To occupy less space in Fig. 1 the number of B.1.620 genomes was reduced down to a representative set of 27, and a phylogeny was inferred using MrBayes v3.2[57] under the HKY+$\Gamma_4$ model of nucleotide substitution[44,56] and rooted on the reference sequence. MCMC was run for 2 million states, sampling every 1000th state and convergence confirmed by checking that effective sample sizes (ESSs) were above 200 for every parameter.

**Reporting summary**. Further information on research design is available in the Nature Research Reporting Summary linked to this article.

## Data availability

SARS-CoV-2 sequence data generated in this study have been deposited in the GISAID database. These sequence data are available under restricted access due to GISAID's Database Access Agreement, access can be obtained by registering an account with GISAID and downloaded via the list of accession used that we provide in the supplementary GISAID acknowledgement table. The processed SARS-CoV-2 genome data in the form of phylogenetic trees are available at https://github.com/evogytis/B.1.620-in-Europe or under Zenodo https://doi.org/10.5281/zenodo.5494346. The SARS-CoV-2 genome data used in this study are available in the GISAID database under accession codes provided in the supplementary acknowledgement table, https://github.com/evogytis/B.1.620-in-Europe, and under Zenodo https://doi.org/10.5281/zenodo.5494346. A list of GISAID accessions for genomes used here, as well as phylogenetic trees used in figures, are available at https://github.com/evogytis/B.1.620-in-Europe or under Zenodo https://doi.org/10.5281/zenodo.5494346. To access sequence data from GISAID one has to register an account with https://www.gisaid.org/, which involves identifying oneself and agreeing to GISAID's Database Access Agreement.

## Code availability

Scripts used to generate figures are available at https://github.com/evogytis/B.1.620-in-Europe or under Zenodo https://doi.org/10.5281/zenodo.5494346. We provide the XML files to perform the Bayesian phylogeographic reconstructions in BEAST 1.10.5[49] as Supplementary Data 2.

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

## Acknowledgements

We gratefully acknowledge the authors from originating laboratories responsible for obtaining the specimens, as well as submitting laboratories where the genome data were generated and shared via GISAID, on which this research is based. An acknowledgement table with GISAID accession IDs of SARS-CoV-2 genomes used here is included. We thank all involved in the collection and processing of SARS-CoV-2 testing and genomic data, as well as associated metadata on individual travel histories. In particular, we would like to thank Marc Noguera Julian, Elisa Martro Catala, Samuel Cordey, Piet Maes, Keith Durkin, Bruno Verhasselt, Lize Cuypers, Lien Cattoir, Veerle Matheeussen, Vincent Enouf, Sylvie van der Werf, Etienne Simon-Lorière, Tobias Schindler, Vladimira Koudelakova, Gabriel Gonzalez, Ariane Düx, Yanthe Nobel, Livia Patrono, Justas Dapkūnas and Andrew J. Tatem. We would like to thank Richard Neher and Kristian G. Andersen for thoughtful discussions. S.L.H. acknowledges support from the Research Foundation - Flanders ('Fonds voor Wetenschappelijk Onderzoek - Vlaanderen,' G0D5117N). B.P. and G.B. acknowledge support from the Internal Fondsen KU Leuven/Internal Funds KU Leuven (Grant No. C14/18/094). G.B. acknowledges support from the Research Foundation - Flanders ('Fonds voor Wetenschappelijk Onderzoek - Vlaanderen,' G0E1420N, G098321N). F.H.L. and T.F.N. were supported by WWF and German Research Council's grant LE1813/14-1 (Great Ape Health in Tropical Africa), Research in CAR took place under permit #098/MRSIT/DIRCAB/CB.20, granted to T.T. by the Ministry of Scientific Research and Technological Innovation. J.S. acknowledges funding from the Dutch Research Council NWO Gravitation 2013 BOO, Institute for Chemical Immunology (ICI; 024.002.009). A.M.J.J.B. acknowledges the support of European Union Horizon 2020 projects BioExcel (823830) and EOSC-Hub (777536) projects. A.A. and C.B. acknowledge the support of the French National Research Institute for Sustainable Development (IRD).

## Author contributions

G.D.—conceptualisation, methodology, formal analysis, investigation, resources, data curation, writing—original draft, visualisation, supervision, project administration, funding acquisition; S.L.H.—formal analysis, data curation, writing—original draft; B.I.P.—formal analysis, data curation, writing—original draft; S.C.-S.—investigation, resources, data curation, writing—review and editing; F.S.N.-S.—investigation, resources; T.B.T.—investigation, resources; T.F.-N.—investigation, resources; U.V.—investigation, resources; M.U.—investigation, resources; F.H.L.—investigation, resources, data curation, writing—review and editing; K.K.—investigation, resources, data curation; C.H.—investigation, resources, data curation; A.W.—investigation, resources, data curation; I.O.—formal analysis, investigation, resources, data curation, writing—review and

editing, project administration, funding acquisition; J.S.—formal analysis, investigation, resources, data curation, writing—original draft, visualisation; K.N.W.—formal analysis, investigation, resources, data curation, writing—original draft, visualisation; A.M.J.J.B.—formal analysis, investigation, resources, data curation, writing—original draft, visualisation; P.M.—resources; S.B.—resources; A.A.—resources; M.F.M.—resources; D.M.D.—resources; C.G.—resources; C.B.—resources; A.S.—investigation, resources; M.G.—resources, data curation, writing—review and editing; M.K.—resources; R.N.—resources; L.R.—resources; G.W.K.—resources; J.K.K.—resources; R.J.—resources; I.N.—resources; Ž.Ž.—resources; D.G.—resources; K.T.—resources; M.N.—resources; E.V.—resources, writing—review and editing; D.Ž.—resources; A.T.—resources; M.S.—resources; M.S.—resources; G.A.—resources; A.A.A.—resources; E.K.L.—resources; J.-C.M.C.—resources; F.M.M.—resources; E.L.L.—resources; P.M.K.—resources; J.-J.M.T.—resources; M.R.D.B.—resources; R.G.E.—resources; M.C.O.A.—resources; A.B.M.—resources; A.B.D.—resources; D.J.—resources, funding acquisition; S.H.—resources; J.O.—resources; M.O.A.—resources; D.N.—resources, funding acquisition; A.P.—resources; C.D.R.—resources; A.V.—resources; R.U.—resources; A.G.—resources; D.Č.—investigation, resources; V.L.—resources; L.Ž.—project administration; L.G.—investigation, resources, funding acquisition; G.B.—conceptualisation, methodology, formal analysis, investigation, resources, data curation, writing—original draft, supervision, project administration.

## Competing interests

The authors declare no competing interests.

## Additional information

Gytis Dudas [1,2,40✉], Samuel L. Hong [3], Barney I. Potter[3], Sébastien Calvignac-Spencer [4,5], Frédéric S. Niatou-Singa[6], Thais B. Tombolomako[6], Terence Fuh-Neba[6], Ulrich Vickos[7,8], Markus Ulrich[4], Fabian H. Leendertz [4], Kamran Khan[9,10,11], Carmen Huber [9], Alexander Watts[9], Ingrida Olendraitė[2,12], Joost Snijder[13], Kim N. Wijnant[13], Alexandre M.J.J. Bonvin [14], Pascale Martres[15], Sylvie Behillil[16,17], Ahidjo Ayouba[18], Martin Foudi Maidadi [19], Dowbiss Meta Djomsi[19], Celestin Godwe[19], Christelle Butel[18], Aistis Šimaitis[20], Miglė Gabrielaitė [21], Monika Katėnaitė[2], Rimvydas Norvilas[2,22], Ligita Raugaitė[2], Giscard Wilfried Koyaweda[23], Jephté Kaleb Kandou[23], Rimvydas Jonikas[24], Inga Nasvytienė[24], Živilė Žemeckienė[24], Dovydas Gečys[25], Kamilė Tamušauskaitė [25], Milda Norkienė [26], Emilija Vasiliūnaitė [26], Danguolė Žiogienė[26], Albertas Timinskas[26], Marius Šukys [24,27], Mantas Šarauskas[24], Gediminas Alzbutas[28], Adrienne Amuri Aziza [29,30], Eddy Kinganda Lusamaki[29,30], Jean-Claude Makangara Cigolo [29,30], Francisca Muyembe Mawete[29,30], Emmanuel Lokilo Lofiko[29], Placide Mbala Kingebeni[29,30], Jean-Jacques Muyembe Tamfum[29,30], Marie Roseline Darnycka Belizaire[31], René Ghislain Essomba[32,33], Marie Claire Okomo Assoumou[32,33], Akenji Blaise Mboringong[32], Alle Baba Dieng[34], Dovilė Juozapaitė [2], Salome Hosch [35], Justino Obama[36], Mitoha Ondo'o Ayekaba[36], Daniel Naumovas [2], Arnoldas Pautienius[37], Clotaire Donatien Rafaï[23], Astra Vitkauskienė[38], Rasa Ugenskienė[24,27], Alma Gedvilaitė [26], Darius Čereškevičius[24,25], Vaiva Lesauskaitė[25], Lukas Žemaitis[25,39], Laimonas Griškevičius[2] & Guy Baele [3,40✉]

[1]Gothenburg Global Biodiversity Centre, Gothenburg, Sweden. [2]Hematology, Oncology and Transfusion Medicine Center, Vilnius University Hospital Santaros Klinikos, Vilnius, Lithuania. [3]Department of Microbiology, Immunology and Transplantation, Rega Institute, KU Leuven, Leuven, Belgium. [4]Epidemiology of Highly Pathogenic Organisms, Robert Koch Institute, 13353 Berlin, Germany. [5]Viral Evolution, Robert Koch Institute, 13353 Berlin, Germany. [6]WWF Central African Republic Programme Office, Dzanga Sangha Protected Areas, BP 1053 Bangui, Central African Republic. [7]Infectious and Tropical Diseases Unit, Department of medicine, Amitié Hospital, Bangui, Central African Republic. [8]Academic Department of Pediatrics, Clinical immunology and vaccinology, Children's Hospital Bambino Gesù, IRCCS, Rome, Italy. [9]BlueDot, Toronto, ON M5J 1A7, Canada. [10]Li Ka Shing Knowledge Institute, St. Michael's Hospital, Toronto, ON M5B 1A6, Canada. [11]Division of Infectious Diseases, Department of Medicine, University of Toronto, Toronto, ON M5S 3H2, Canada. [12]Division of Virology, Department of Pathology, University of Cambridge, Addenbrooke's Hospital Lab, CB2 2QQ Cambridge, UK. [13]Biomolecular Mass Spectrometry and Proteomics, Bijvoet Center for

Biomolecular Research and Utrecht Institute of Pharmaceutical Sciences, Utrecht University, Padualaan 8, 3584 CH Utrecht, The Netherlands. [14]Bijvoet Centre for Biomolecular Research, Faculty of Science - Chemistry, Utrecht University, Padualaan 8, 3584 CH Utrecht, The Netherlands. [15]Microbiology, Centre Hospitalier René Dubos, Cergy Pontoise, France. [16]Molecular Genetics of RNA viruses, CNRS UMR 3569, Université de Paris, Institut Pasteur, Paris, France. [17]National Reference Center for Respiratory Viruses, Institut Pasteur, Paris, France. [18]TransVIHMI, Université de Montpellier, IRD, INSERM, 911 Avenue Agropolis, 34394 Montpellier cedex, France. [19]Centre de Recherches sur les Maladies Émergentes, Ré-émergentes et la Médecine Nucléaire, Institut de Recherches Médicales et D'études des Plantes Médicinales, Yaoundé, Cameroon. [20]The Office of the Government of the Republic of Lithuania, Vilnius, Lithuania. [21]Center for Genomic Medicine, Rigshospitalet, Copenhagen, Denmark. [22]Department of Experimental, Preventive and Clinical Medicine, State Research Institute Centre for Innovative Medicine, Vilnius, Lithuania. [23]Le Laboratoire National de Biologie Clinique et de Santé Publique (LNBCSP), Bangui, Central African Republic. [24]Department of Genetics and Molecular Medicine, Hospital of Lithuanian University of Health Sciences Kauno Klinikos, Kaunas, Lithuania. [25]Institute of Cardiology, Lithuanian University of Health Sciences, Kaunas, Lithuania. [26]Institute of Biotechnology, Life Sciences Center, Vilnius University, Vilnius, Lithuania. [27]Department of Genetics and Molecular Medicine, Lithuanian University of Health Sciences, Kaunas, Lithuania. [28]Institute for Digestive Research, Lithuanian University of Health Sciences, Kaunas, Lithuania. [29]National Institute for Biomedical Research (INRB), Avenue De la Democratie (Ex Huileries), BP 1197 Kinshasa-Gombe, Democratic Republic of the Congo. [30]University of Kinshasa (UNIKIN), BP 127 Kinshasa XI, Democratic Republic of the Congo. [31]World Health Organization, Central African Republic Office, Bangui, Central African Republic. [32]National Public Health Laboratory, Ministry of Public Health, Yaoundé, Cameroon. [33]Faculty of Medicine and Biomedical Sciences, University of Yaoundé I, Yaoundé, Cameroon. [34]World Health Organization, Cameroon Office, Yaoundé, Cameroon. [35]Swiss Tropical and Public Health Institute, Basel, Switzerland. [36]Ministry of Health and Social Welfare, Malabo, Equatorial Guinea. [37]Institute of Microbiology and Virology, Lithuanian University of Health Sciences, Kaunas, Lithuania. [38]Department of Laboratory Medicine, Lithuanian University of Health Sciences, Kaunas, Lithuania. [39]National Public Health Surveillance Laboratory, Vilnius, Lithuania. [40]These authors contributed equally: Gytis Dudas, Guy Baele. ✉email: gytisdudas@gmail.com; guy.baele@kuleuven.be

