## [Peer Review File · Nature Communications]

REVIEWER COMMENTS

Reviewer #1 (Remarks to the Author):

This study by Dudas et al. describes the emergence of a SARS-CoV-2 lineage carrying many mutations and deletions in the spike protein that are shared with variants of concerns. The lineage designated B.1.620 first described in Lithuania after intense efforts has now been found in several European countries and in central Africa. New lineages will constantly appear and disappear so the discovery of a lineage per se may be potentially interesting to some but may not merit huge interest to general readers unless it is a variant of concern or a variant under investigation. The finding that this lineage emerged from travel (as suggested in the title) is not exactly novel as that is how the entire pandemic has emerged. Nevertheless, this is a very well written and described study that articulates how a lineage that bears E484K, S477N and numerous B.1.1.7 and B.1.351-like deletions can potentially originate and essentially evolve and spread cryptically. However, since submission of the manuscript the number of B.1.620 designated genomes has increased and there are some new genomes from Africa (Gabon, CAR, Ghana, Cameroon, Equatorial Guinea). Addition of these genomes should help illuminate even further the historical pathway of this lineage and aid in describing the potential source of this lineage.

Comments

- While I understand that more and more genomes are deposited each day and the authors can't keep refining their analysis. I do believe that including more genomes would be a strength here as the authors analysis are based on a relatively small number (238) of B.1.620 genomes and especially given the authors conclusions. There are now over 530 genomes available on GISAID (twice of that studied here) and 3 of the earliest described B.1.620 (assuming the pango lineage is actually correctly listed) are from Switzerland (EPI_ISL_1598004, EPI_ISL_1658311 and EPI_ISL_1657740) with collection dates in early December. I don't see these genomes as part of this study and wonder how the addition of these genomes would help reconstruct the order of mutations and the genesis of this lineage. It would be interesting to examine the mutational pattern of these early genomes and compare them as illustrated in figure 1.
- There are a number of genomes from CAR that don't have deposited dates in GISAID. What dates did the authors use for their phylogeographic analysis?
- The likelihood that this lineage arose in Africa is based on a very few small number of genomes from Africa and given the unevenness of genomic sampling efforts I think more caution should be exercised when trying to assess the source of this lineage especially in light on the many papers and commentaries which state the difficulties in using phylogenetics to pinpoint any geographical source for SARS-CoV-2 lineages. I appreciate how sequencing infected travelers can be used a proxy but from the collected B.1.620 genomes only 7 are from travelers returning from Cameroon whereas six were sampled in the CAR. I worry about the overall conclusions when based on such a small number. For instance, the statement that B.1.620 likely circulated at high frequency in Central Africa is not supported by the raw numbers even in the absence of routine genomic surveillance. Moreover, the long branch that connects this lineage to its ancestor illustrates the long unsampled period which may even originated outside of Central Africa.
- Were any sites masked in the alignment such as those listed as homoplasies?
- I congratulate the authors for their transparencies and making the scripts and some files available via github but the xml files and travel related files are not provided on the github. Can these also be included.
- The authors need to beware of stigma that may be associated with the evidence that B.1.620 originated in Central Africa (Even if this is true) and how this may hamper efforts by African scientist to release data into the public. Hence, the need to be precise and accurate with reporting and all limitations should be clearly stated.

- If this lineage is likely to escape Ab-mediated immunity why do the authors think that we haven't seen more cases of B.1.620. Most of the cases in Europe appear due to sporadic outbreaks. I guess B.1.1.7 was also taking off at this time and this variant could not out compete that lineage.
- The authors should explicitly state in the results the estimated tMRCA and 95% HPD. Although one can see this from Figure 4. I would like to see what the 95% HPD are.

Reviewer #2 (Remarks to the Author):

This interesting paper is about a particular SARS-CoV-2 variant that is prevalent in Lithuania (Spring 2021), the mutations it carries and its likely inferred temporal and geographic origin. In a way the results are not overly surprising - a 'new variant' was imported from an under sampled location, but the story of how this was detected, the description of the analysis methods required, and the subsequent result of the 'new variant' in the population make a nice study and show what can be done and understood about the pandemic with genomics even without ultra-dense sampling.

Minor comments

Introduction -

The introduction section nicely introduces the variants and puts the history of why certain variants might be considered of interest, under investigation or of concern into context. But please check that you've defined (and distinguished) VOI, VUI and VOC in properly. I think what you've written is OK but something might have gotten lost in the edit, and you don't mention VOI until the 3rd paragraph.

Results -

Can you summarise/headline at the start of the paragraph some of the important spike protein mutations in B.1.620 - e.g. "Lineage B.1.620 attracted our attention due to large numbers of unique mutations, i.e. its genomes are distantly related to available references ($\geq X$ nucleotide substitutions)". And "Despite sharing multiple mutations and deletions with known VOCs, including spike protein substitutions S477N, E484K, spike protein deletion at position 69-70" - or whatever few (3 is usually a good number) you think are worth highlighting. The detailed section you have later on starting line 204 is good though.

Travel history and phylogeography sections - to me these are the heart of the paper, and are well described with use of appropriate methods. Even the phylogeographic location inference, which can be problematic when especially in biased or under sampling, has been performed in ways that mitigate this as much as possible and is described with suitable caveats.

Discussion -

Interesting about the lack of civil air travel to Mali - suggest that you insert the word 'civil' on line 326 "despite higher civil air passenger". When you put it like that, it highlights that the civil air passenger data is not the only means of air transport as you note, there could be military or maybe just other private air travel that is not captured in the air passenger data; it might be better to just re-word to emphasise this point rather than speculate.

Methods / General & Figure 4

You rightly point out the long branch leading to the ancestral node of B.1.620 - can you put an indication on that branch to highlight on this figure that this is the long branch you are talking about

(yes there will be readers who don't look at trees all day every day..). But also you can see from that figure (looking carefully), there are other quite long branches also in Africa, which are understandable given the much lower and later in time amount of sequencing, but are these also a result of you including the random 150 sequences from nextstrain ? And if you chose a different set of 150 sequences (or other background sequences) would you get a less long branch ? Not that I'm suggesting that you should redo all the bayesian trees multiple times or anything, but you might want to comment somewhere about this.

Reviewer #3 (Remarks to the Author):

Dudas et al. present a detailed description of the emergence and spread of the B.1.620 lineage of SARS-CoV-2 in Lithuania and other European countries. Their phylogenetic analysis establishes local transmission of B.1.620 in Europe and that the variant was likely imported several times into Europe from Cameroon or elsewhere in Central Africa.

My one major criticism is that the analysis does not directly address the epidemiological relevance of B.1.620 in the context of the current situation where multiple variants with increased transmissibility and/or with antigenic escape mutations. Establishing local transmission and determining the geographic origin of a variant is important, but its of secondary importance relative to determining a variants transmission potential relative to other variants. Can the authors make any inference from their phylogenetic analysis or the genomic sampling data about the relative fitness and transmissibility of B.1.620 compared to other variants of interest? Is the percentage of B.1.620 confirmed cases growing over time? Is there evidence that B.1.620 is spreading faster in more immunized or vaccinated populations?

Minor points:

Lines 220-221: "There is preliminary evidence that B.1.620 can infect fully vaccinated individuals." - It might be worth stating what is known about these breakthrough cases. Were they discovered through testing/surveillance or did they result in severe cases with hospitalizations/deaths? Since any variant can in theory infect vaccinated individuals without cause for concern regarding onwards transmission or severe disease outcomes, knowing these details would help the reader assess the antigenic relevance of B.1.620.

Line 325-326: "If B.1.620 were to be introduced to Mali via land routes it would first need to rise to high frequency in countries between Central Africa and Mali, at very least Nigeria and Niger." This is a bizarre argument to make and undermined by the much more plausible scenario proposed in very next sentence about direct travel from Cameroon to Mali.

“Travel-driven emergence and spread of SARS-CoV-2 lineage B.1.620 with multiple VOC-like mutations and deletions in Europe”: point-by-point responses to reviewer comments

July 29, 2021

For ease-of-reading, we have divided the reviewers comments into numbered parts and replied to each point separately. Reviewer’s comments are shown in italics and between quotation marks.

Reviewer 1

1.1

“ This study by Dudas et al. describes the emergence of a SARS-CoV-2 lineage carrying many mutations and deletions in the spike protein that are shared with variants of concerns. The lineage designated B.1.620 first described in Lithuania after intense efforts has now been found in several European countries and in central Africa. New lineages will constantly appear and disappear so the discovery of a lineage per se may be potentially interesting to some but may not merit huge interest to general readers unless it is a variant of concern or a variant under investigation. The finding that this lineage emerged from travel (as suggested in the title) is not exactly novel as that is how the entire pandemic has emerged. Nevertheless, this is a very well written and described study that articulates how a lineage that bears E484K, S477N and numerous B.1.1.7 and B.1.351-like deletions can potentially originate and essentially evolve and spread cryptically. However, since submission of the manuscript the number of B.1.620 designated genomes has increased and there are some new genomes from Africa (Gabon, CAR, Ghana, Cameroon, Equatorial Guinea). Addition of these genomes should help illuminate even further the historical pathway of this lineage and aid in describing the potential source of this lineage.

Comments - While I understand that more and more genomes are deposited each day and the authors can’t keep refining their analysis. I do believe that including more genomes would be a strength here as the authors’ analysis are based on a relatively small number (238) of B.1.620 genomes and especially given the authors conclusions. There are now over 530 genomes available on GISAID (twice of that studied here) and 3 of the earliest described B.1.620 (assuming the pango lineage is actually correctly listed) are from

Switzerland (EPI_ISL_1598004, EPI_ISL_1658311 and EPI_ISL_1657740) with collection dates in early December. I don't see these genomes as part of this study and wonder how the addition of these genomes would help reconstruct the order of mutations and the genesis of this lineage. It would be interesting to examine the mutational pattern of these early genomes and compare them as illustrated in Figure 1."

Response: We thank the Reviewer for this positive assessment of our work. Indeed, as with many SARS-CoV-2 analyses, the number of available genomes increases during the manuscript review process and some of these may be the result of retrospective sequencing efforts. Nevertheless, we have kept an eye on new B.1.620 sequences on GISAID during the review process and found nothing that would markedly alter the story presented in our manuscript, which focuses on the African origin of lineage B.1.620 and its subsequent spread to and within Europe.

There are only three additional B.1.620 sequences from Africa with precise collection dates beyond those used in the manuscript, i.e. one travel case in Ghana without mention of country of origin and two cases from Gabon. The three putative older B.1.620 sequences the reviewer mentions are no longer classified as B.1.620 on GISAID, but other new B.1.620 sequences have been uploaded since the submission of our original manuscript. One of these is a French genome that predates the oldest one used in our analyses by one day (February 25, 2021) and another one of these is a genome from Cameroon that wasn't used in our analyses that is now the oldest known Cameroonian sequence (February 10, 2021).

As the Reviewer mentions, more and more genomes are deposited into GISAID each day, and this has indeed also been the case for lineage B.1.620. For example, a large collection of more than 320 South Korean B.1.620 genomes have been uploaded to GISAID from a substantial outbreak there. Interestingly, one of these genomes – with a collection date of March 27, 2021 – constitutes a travel case from Kenya, a country from which no B.1.620 genomes are available to this date. We have updated Figure 3 ('Known locations and travel history of B.1.620 cases') in the main text with this information. During the review process additional B.1.620 from Republic of the Congo appeared on GISAID which we simply could not include in revised analyses because producing such high quality datasets takes an increasingly long time, does not alter any of our findings in any substantial way and phylogeographic analyses take up inordinate amounts of time given the number of B.1.620 sequences available.

We agree on the importance of providing analyses that are as up-to-date as possible, and have hence updated our data set with B.1.620 genomes that have become available during our original study period, i.e. from the origin of lineage B.1.620 until the first week of May, and hence including the travel case from Kenya to South Korea. Among these are also a collection of 37 genomes from the Central African Republic (CAR) for which only the sampling year (2021) is known; we included these genomes in our updated analysis and performed tip-date sampling to estimate their collection dates. Overall, this enabled us to increase the number of genomes from CAR in our data set from 6 (provided by our collaborators for the original submission of our manuscript) to 49 for the analysis in this revised version of our manuscript.

Our results on this updated data set now put the estimated origin of B.1.620 in the Central African Republic, leaving our story of a Central African origin intact with shifts from Cameroon (our very first draft on medRxiv) to CAR (updated version submitted to Nature Communications) and now a mix of the two with CAR being the more likely. In order to offer a means of

comparison for the impact of retrospective genome sequencing, we have moved our original result into Supplementary Materials (i.e. Supplementary Figures S9 and S10) and have updated our original Figure 4 to show our new results.

1.2

“ There are a number of genomes from CAR that don’t have deposited dates in GISAID. What dates did the authors use for their phylogeographic analysis? ”

Response: We know from the GISAID entries that these genomes had a sampling date in 2021. We made use of a tip-date sampling procedure available in BEAST to integrate out a time frame from January 1st until June, 2021, i.e. the registered date of submission to GISAID for these genomes. We refer to http://beast.community/tip_date_sampling for more information on this procedure, which adds half of the uncertainty period to the assumed sampling date in the BEAST XML file (i.e. January 1st, 2021, for these CAR genomes) and constructs a uniform window of roughly 4 months around this new value. We now mention this in more detail in the revised manuscript and provide the BEAST XML files in Supplementary Material.

1.3

“ The likelihood that this lineage arose in Africa is based on a very few small number of genomes from Africa and given the unevenness of genomic sampling efforts I think more caution should be exercised when trying to assess the source of this lineage especially in light on the many papers and commentaries which state the difficulties in using phylogenetics to pinpoint any geographical source for SARS-CoV-2 lineages. I appreciate how sequencing infected travelers can be used a proxy but from the collected B.1.620 genomes only 7 are from travelers returning from Cameroon whereas six were sampled in the CAR. I worry about the overall conclusions when based on such a small number. For instance, the statement that B.1.620 likely circulated at high frequency in Central Africa is not supported by the raw numbers even in the absence of routine genomic surveillance. Moreover, the long branch that connects this lineage to its ancestor illustrates the long unsampled period which may even originated outside of Central Africa.”

Response: We have to respectfully disagree with the Reviewer here. We made sure to refer to the region of Central Africa as the immediate, not ultimate, origin of lineage B.1.620, precisely because of the long branch the Reviewer mentions. To support this immediate Central African origin claim we have used a total of 49 genomes from the Central African Republic (instead of the six that were available at the time when we posted a preprint on medRxiv) occupying phylogenetic positions exactly expected of an immediate origin location combined with travel cases from Cameroon that intermingle with B.1.620 diversity found in Europe. Recent sequencing efforts carried out in African countries, including countries that are geographically close to Central Africa, have been substantial and to date have not identified basal B.1.620 genotypes or their antecedents, nor evidence that B.1.620 circulates at a high frequency anywhere else. The sole country in the world where B.1.620 is the dominant circulating variant is Central African Republic. Thus, in the absence of evidence to the contrary, we have shown that lineage B.1.620 in Europe primarily came from Central Africa beyond

reasonable doubt using multiple lines of evidence without speculation about this lineage's ultimate origins which, we agree with the Reviewer, remain unknown.

1.4

“ Were any sites masked in the alignment such as those listed as homoplasies?”

Response: This is a very important question since one would expect that highly convergent evolution would lead to inferring incorrect topologies. The combination of three key mutations (a synonymous change at site 15324, S:E484K, and S:T1027I) and one deletion (ORF1a: Δ 3675/3677) appear sufficient to identify B.1.620 as a relative to B.1.619 (also of presumed Central African origin) with the synonymous change at site 15324 placing both of these lineages as close relatives to early Cameroonian sequences (see Figure 1 in the main text). One of the initial hypotheses we pursued upon identification of B.1.620 is that it might be of recombinant origin, but whenever we looked into potential donor lineages they turned out to have SNPs within putative recombination tracts that weren't present in B.1.620.

In terms of masking, we ignored the first 100 and the last 50 nucleotides of all sequences displayed in Figure 1 and Supplementary Figure S4, but this was a visual alteration only and phylogenetic trees were inferred using the full alignment. We also manually removed clusters of mutations near the ends of some B.1.620 genomes generated for Lithuania by the ECDC, as well as in other sequences that seemed suspect. We have now clarified both of these in the methods section, as follows:

“Some sequences had clusters of SNPs different from the reference at the ends of the genome, particularly the 5' end. In such cases the ends of the genomes were trimmed to exclude these regions of likely sequencing or assembly error.”

and

“The first 100 and the last 50 nucleotides are not included in the figure but were used to infer the phylogeny.”

1.5

“ I congratulate the authors for their transparencies and making the scripts and some files available via GitHub but the XML files and travel-related files are not provided on the GitHub. Can these also be included?”

Response: We thank the Reviewer for their appreciation. The individual travel histories are part of the BEAST XML files, as this information is exploited by the travel history-aware phylogeographic reconstruction (Lemey et al., 2020). We now provide these XML files and also clearly mention this in the revised manuscript (in the Data Availability section).

1.6

“ The authors need to beware of stigma that may be associated with the evidence that B.1.620 originated in Central Africa (even if this is true) and how this may hamper efforts by African scientists to release data into the public. Hence, the authors need to be precise

and accurate with reporting and all limitations should be clearly stated. ”

Response: We are most certainly aware of the great care that must be taken to not point fingers in our work. This is why we refer to the Central Africa region as a whole as the immediate, not ultimate, source of lineage B.1.620, rather than individual countries where the lineage was identified and, as we have stated in response 1.3 to the Reviewer, the entirety of the data at hand consistently points to this. Additionally, we would like to point out that we place blame in the manuscript where blame is due, namely the global inequalities that have led to a lack of vaccines being available in poor countries, which increases the chances of lineages like B.1.620 arising, and lack of sequencing capacity, which prevented the early detection of lineage B.1.620 until it spread elsewhere unnoticed.

1.7

“ If this lineage is likely to escape Ab-mediated immunity why do the authors think that we haven’t seen more cases of B.1.620. Most of the cases in Europe appear due to sporadic outbreaks. I guess B.1.1.7 was also taking off at this time and this variant could not out compete that lineage.”

Response: The Reviewer makes a very fair and excellent point. However, we disagree that B.1.620 in Europe can be said to only cause sporadic outbreaks. Granted, in European countries where B.1.620 was introduced multiple times each introduction had highly variable success but what looked like sporadic outbreaks in March/April 2021 in Lithuania, Portugal, France, Germany, Switzerland and Belgium are now very persistent transmission chains circulating as late as June 2021 in multiple countries. We believe that a combination of stricter control measures brought about by B.1.1.7, increasingly vaccinated populations and spring in Europe may have all contributed to mitigating the transmission of B.1.620 but not stopping it entirely. We also disagree that escaping Ab-mediated immunity necessarily guarantees a marked transmission advantage, for example B.1.351, a very potent Ab-evading lineage, has not fared very well in Europe.

We have added a new figure (Supplementary Figure S6) to the manuscript (also in response to comment 1 by Reviewer 3) that shows relative frequencies of lineages B.1.620 and B.1.1.7 across five European countries that saw most of B.1.620 transmission: Lithuania, Germany, France, Switzerland, and Belgium. This figure also shows the cases and proportion of population receiving at least one vaccine dose for epidemiological context. Based on this figure it seems that increasingly vaccinated European populations have not exerted much selective pressure on B.1.1.7 until quite recently, arguably May.

1.8

“ The authors should explicitly state in the results the estimated tMRCA and 95% HPD. Although one can see this from Figure 4, I would like to see what the 95% HPD are.”

Response: We now include the estimated tMRCA date and 95% HPD intervals in Figure 4 for continent-level analysis – which includes numerous lineages – and for the country-level analysis of B.1.620.

Reviewer 2

2.1

“ This interesting paper is about a particular SARS-CoV-2 variant that is prevalent in Lithuania (Spring 2021), the mutations it carries and its likely inferred temporal and geographic origin. In a way the results are not overly surprising - a ‘new variant’ was imported from an under sampled location, but the story of how this was detected, the description of the analysis methods required, and the subsequent result of the ‘new variant’ in the population make a nice study and show what can be done and understood about the pandemic with genomics even without ultra-dense sampling. ”

Response: We thank the Reviewer for the appreciation of our work.

2.2

“ Minor comments:

The introduction section nicely introduces the variants and puts the history of why certain variants might be considered of interest, under investigation or of concern into context. But please check that you’ve defined (and distinguished) VOI, VUI and VOC in properly. I think what you’ve written is OK but something might have gotten lost in the edit, and you don’t mention VOI until the 3rd paragraph. ”

Response:

Our apologies for the omission. We have included a third sentence to the first paragraph of the introduction that reads:

“An even broader category termed variant of interest (VOI) encompasses lineages that are suspected to have an altered phenotype implied by their mutation profile.”

2.3

“ Can you summarise/headline at the start of the paragraph some of the important spike protein mutations in B.1.620 - e.g. “Lineage B.1.620 attracted our attention due to large numbers of unique mutations, i.e. its genomes are distantly related to available references ($\geq X$ nucleotide substitutions)”. And “Despite sharing multiple mutations and deletions with known VOCs, including spike protein substitutions S477N, E484K, spike protein deletion at position 69-70” - or whatever few (3 is usually a good number) you think are worth highlighting. The detailed section you have later on starting line 204 is good though. ”

Response: Thank you for the suggestions, the alterations do make it much easier for the reader.

“Lineage B.1.620 attracted our attention due to large numbers of unique mutations in Lithuanian B.1.620 genomes in nextclade analyses (its genomes are 18 mutations away from nearest relatives and 26 from reference strain Wuhan-Hu-1), ...”

and

“Despite sharing multiple mutations and deletions with known VOCs (most prominently HV69/70Δ, LLA241/243Δ, S477N, E484K, and P681H), ...”

2.4

“ Travel history and phylogeography sections - to me these are the heart of the paper, and are well described with use of appropriate methods. Even the phylogeographic location inference, which can be problematic when especially in biased or under sampling, has been performed in ways that mitigate this as much as possible and is described with suitable caveats.”

Response: We thank the Reviewer for appreciating our use of state-of-the-art inference approaches.

2.5

“ Interesting about the lack of civil air travel to Mali - suggest that you insert the word ‘civil’ on line 326 “despite higher civil air passenger”. When you put it like that, it highlights that the civil air passenger data is not the only means of air transport as you note, there could be military or maybe just other private air travel that is not captured in the air passenger data; it might be better to just re-word to emphasise this point rather than speculate. ”

Response: Thank you for the suggestion, we have replaced the two sentences in question with one that reads (note that the sentences were also altered in response 3 to Reviewer 3):

“We consider the introduction of B.1.620 from Central Africa to Mali via land routes improbable, since outbreaks caused by B.1.620 have not been observed in Niger and Nigeria, the countries separating the region from Mali. The lack of any B.1.620 genomes from Nigeria in particular, one of the leaders in SARS-CoV-2 genome sequencing on the continent to date, despite higher civil air passenger volumes (Figure 5) suggests other means of long-distance travel between Central Africa and Mali (EU, 2016, 2013).”

2.6

“ You rightly point out the long branch leading to the ancestral node of B.1.620 - can you put an indication on that branch to highlight on this figure that this is the long branch you are talking about (yes there will be readers who don’t look at trees all day every day ...). But also you can see from that figure (looking carefully), there are other quite long branches also in Africa, which are understandable given the much lower and later in time amount of sequencing, but are these also a result of you including the random 150 sequences from nextstrain? And if you chose a different set of 150 sequences (or other background sequences) would you get a less long branch ? Not that I’m suggesting that you should redo all the Bayesian trees multiple times or anything, but you might want to comment somewhere about this. ”

Response: This is something we worried about ourselves. A lot of effort not apparent in the manuscript was spent on looking for relatives of B.1.620 closer than B.1.619, involving

customised BLAST databases and variously masked B.1.620-like queries. At this point it seems unlikely that the long branch will be broken up by new sequences. We have described this BLAST-based search procedure in methods.

Reviewer 3

3.1

“ Dudas et al. present a detailed description of the emergence and spread of the B.1.620 lineage of SARS-CoV-2 in Lithuania and other European countries. Their phylogenetic analysis establishes local transmission of B.1.620 in Europe and that the variant was likely imported several times into Europe from Cameroon or elsewhere in Central Africa.

My one major criticism is that the analysis does not directly address the epidemiological relevance of B.1.620 in the context of the current situation where multiple variants with increased transmissibility and/or with antigenic escape mutations. Establishing local transmission and determining the geographic origin of a variant is important, but it's of secondary importance relative to determining a variant's transmission potential relative to other variants. Can the authors make any inference from their phylogenetic analysis or the genomic sampling data about the relative fitness and transmissibility of B.1.620 compared to other variants of interest? Is the percentage of B.1.620 confirmed cases growing over time? Is there evidence that B.1.620 is spreading faster in more immunized or vaccinated populations? ”

Response: We agree that demonstrating that B.1.620 is different from other co-circulating lineages is important. We are generally of the opinion that associations between individual mutations and altered antigenic profiles are quite robust, unlike inherent transmission advantage that continues to be difficult to gauge from mutations alone until it's too late. On the basis of the E484K mutation alone we believe the question is not whether B.1.620 has an altered antigenic profile but what is its magnitude given all the other VOC-like mutations and deletions it has. To address this we have added an additional figure to the manuscript (Supplementary Figure S6, also in response to comment 7 by Reviewer 1) that tracks the cases and vaccinations in five European countries that saw the most B.1.620 transmission (Lithuania, Germany, France, Switzerland, and Belgium) as well as the proportion of cases in each of these countries that are caused by B.1.1.7, the predominant variant across much of Europe in spring, and B.1.620.

As we mentioned in response 7 to Reviewer 1 we believe that epidemiological circumstances that give antigenically drifted SARS-CoV-2 lineages an edge over "wild type"-like genotypes are probably recent and so few data points are available to infer the selection coefficient with any precision, particularly when cases are low in the northern hemisphere. We nonetheless observe that B.1.620 does not appear to respond to either increasingly vaccinated populations nor unfavourable climatic conditions unlike B.1.1.7 which is far more prevalent yet has been in decline across all five countries.

3.2

“ Lines 220-221: “There is preliminary evidence that B.1.620 can infect fully vaccinated individuals.” - It might be worth stating what is known about these breakthrough cases. Were they discovered through testing/surveillance or did they result in severe cases with hospitalizations/deaths? Since any variant can in theory infect vaccinated individuals without cause for concern regarding onwards transmission or severe disease outcomes, knowing these details would help the reader assess the antigenic relevance of B.1.620. ”

Response: While we are curious about this ourselves we cannot go beyond public statements made by employees of the National Public Health Centre of Lithuania to media outlets. The comments concerned an outbreak investigation where seven cases at an elderly care facility were all said to be asymptomatic. We were able to extract additional information from sequencing indications. As a result we reorganised the results section and moved the sentence in question to its own paragraph:

“While only limited empirical data are available, they seem to agree with the expectation that B.1.620 is likely to be antigenically drifted relative to primary genotypes. A report presented to the Lithuanian government on May 22, 2021 (Šimaitis, 2021) indicated that amongst 101 sequenced B.1.620 cases at the time, 13 were infections in fully vaccinated individuals, five of whom were younger than 57 years old. Though not systematised properly, sequencing indications for a substantial number of Lithuanian SARS-CoV-2 genomes were available, of which 213 were “positive PCR at least two weeks after second dose of vaccine”, of which 195 were B.1.1.7 and 12 were B.1.620. Since detection of the first B.1.620 case on March 15 in Lithuania approximately 10 000 SARS-CoV-2 genomes were sequenced to date, 9 251 of which were B.1.1.7 and 248 of which were B.1.620. Thus B.1.620 is found 2.4 times more often in vaccine breakthrough cases compared to its population prevalence, whereas for B.1.1.7 this enrichment is only 1.05-fold. Similarly, the frequency of B.1.620 across five most affected European countries (Lithuania, Germany, Switzerland, France, and Belgium) appears relatively stable though at a low level, unlike B.1.1.7 which has been in noticeable decline since April-May (Supplementary Figure S6), presumably on account of increasing vaccination rates and improving weather in Europe.”

3.3

“ Line 325-326: “If B.1.620 were to be introduced to Mali via land routes it would first need to rise to high frequency in countries between Central Africa and Mali, at very least Nigeria and Niger.” This is a bizarre argument to make and undermined by the much more plausible scenario proposed in very next sentence about direct travel from Cameroon to Mali.”

Response: Yes, apologies for the confusion here. It was very much our intention to imply that long-range travel has to have taken place to explain cases in Mali because, as we stated in the sentence following this one, Nigeria’s genomic surveillance programme is too good to have missed a local outbreak of B.1.620. We have rephrased the two neighbouring sentences accordingly (this sentence was also altered in response 5 to Reviewer 2):

“We consider the introduction of B.1.620 from Central Africa to Mali via land routes improbable, since outbreaks caused by B.1.620 have not been observed in Niger and Nigeria, the countries separating the region from Mali. The lack of any B.1.620 genomes from Nigeria in particular, one of the leaders in SARS-CoV-2 genome sequencing on the continent to date, despite higher civil air passenger volumes (Figure 5) suggests other means of long distance travel between Central Africa and Mali (EU, 2016, 2013).”

REVIEWERS' COMMENTS

Reviewer #1 (Remarks to the Author):

The authors have addressed all of my concerns in a robust manner. I have no further comments.

Reviewer #2 (Remarks to the Author):

Thank you for considering my comments. The revisions you have made in response to my comments are good. I've also reviewed the comments of the other reviewers and your subsequent replies; the changes you have made for those also seem good.

In particular the issue of including the more recently deposited sequence data in the analysis - I think what matters here is if the new sequences would be genetically close to your existing sequence set and substantially change key TMRCAs (not just narrow the confidence intervals of the MRCAs). So what you have done, in updating the dataset / analysis slightly seems suitable.

Also, the point about the long branches, and trying to find similar sequences to fill in as much as possible using customised BLAST - this is good and what you've written in the methods explains it suitably.

XMLs - good that you have included the xmls on the GitHub and I notice that you have not included the actual sequence data inside them (which would break the GISAID terms). Do you want to include the pdf GISAID acknowledgement table in the figures folder on GitHub as well ?

Reviewer #3 (Remarks to the Author):

The authors have appropriately responded to my previous concerns. I have no other comments at this time.

“Travel-driven emergence and spread of SARS-CoV-2 lineage B.1.620 with multiple VOC-like mutations and deletions in Europe”: point-by-point responses to reviewer comments

September 8, 2021

For ease-of-reading, we have divided the reviewers comments into numbered parts and replied to each point separately. Reviewer’s comments are shown in italics and between quotation marks.

Reviewer 1

“ The authors have addressed all of my concerns in a robust manner. I have no further comments. ”

Response: Thank you.

Reviewer 2

“

In particular the issue of including the more recently deposited sequence data in the analysis - I think what matters here is if the new sequences would be genetically close to your existing sequence set and substantially change key TMRCAs (not just narrow the confidence intervals of the MRCAs). So what you have done, in updating the dataset / analysis slightly seems suitable.

Also, the point about the long branches, and trying to find similar sequences to fill in as much as possible using customised BLAST - this is good and what you've written in the methods explains it suitably.

XMLs - good that you have included the xmls on the GitHub and I notice that you have not included the actual sequence data inside them (which would break the GISAID terms). Do you want to include the pdf GISAID acknowledgement table in the figures folder on GitHub as well ? *Thank you for considering my comments. The revisions you have made in response to my comments are good. I've also reviewed the comments of the other reviewers and your subsequent replies; the changes you have made for those also seem good.*

In particular the issue of including the more recently deposited sequence data in the analysis - I think what matters here is if the new sequences would be genetically close to your existing sequence set and substantially change key TMRCAs (not just narrow the confidence intervals of the MRCAs). So what you have done, in updating the dataset / analysis slightly seems suitable.

Also, the point about the long branches, and trying to find similar sequences to fill in as much as possible using customised BLAST - this is good and what you've written in the methods explains it suitably.

XMLs - good that you have included the xmls on the GitHub and I notice that you have not included the actual sequence data inside them (which would break the GISAID terms). Do you want to include the pdf GISAID acknowledgement table in the figures folder on GitHub as well ? ”

Response: Thank you. We have included the GISAID acknowledgment table under https://github.com/evogytis/B.1.620-in-Europe/blob/main/data/acknowledgment_table/gisaid_hcov-19_acknowledgement_table_2021_07_29_10.pdf on 2021 August 2, but we have added an additional copy to the figures folder on GitHub too, per request.

Reviewer 3

“ The authors have appropriately responded to my previous concerns. I have no other comments at this time.”

Response: Thank you.